# Macrophage Migration Inhibitory Factor (MIF) and D-Dopachrome Tautomerase (DDT): Pathways to Tumorigenesis and Therapeutic Opportunities

**DOI:** 10.3390/ijms25094849

**Published:** 2024-04-29

**Authors:** Caroline Naomi Valdez, Gabriela Athziri Sánchez-Zuno, Richard Bucala, Thuy T. Tran

**Affiliations:** 1School of Medicine, Yale University, 333 Cedar St., New Haven, CT 06510, USA; caroline.valdez@yale.edu (C.N.V.); richard.bucala@yale.edu (R.B.); 2Section of Rheumatology, Allergy and Immunology, Department of Internal Medicine, Yale University, 333 Cedar St., New Haven, CT 06510, USA; gabriela.sanchez-zuno@yale.edu; 3Yale Cancer Center, Yale University, 333 Cedar St., New Haven, CT 06510, USA; 4Section of Medical Oncology, Department of Internal Medicine, Yale University, 333 Cedar St., New Haven, CT 06510, USA

**Keywords:** Macrophage Migratory Inhibition Factor, MIF, D-dopachrome tautomerase, DDT, cytokines, cancer, oncology

## Abstract

Discovered as inflammatory cytokines, MIF and DDT exhibit widespread expression and have emerged as critical mediators in the response to infection, inflammation, and more recently, in cancer. In this comprehensive review, we provide details on their structures, binding partners, regulatory mechanisms, and roles in cancer. We also elaborate on their significant impact in driving tumorigenesis across various cancer types, supported by extensive in vitro, in vivo, bioinformatic, and clinical studies. To date, only a limited number of clinical trials have explored MIF as a therapeutic target in cancer patients, and DDT has not been evaluated. The ongoing pursuit of optimal strategies for targeting MIF and DDT highlights their potential as promising antitumor candidates. Dual inhibition of MIF and DDT may allow for the most effective suppression of canonical and non-canonical signaling pathways, warranting further investigations and clinical exploration.

## 1. Introduction

Macrophage Migration Inhibitory Factor (MIF) was first described in the 1960s as an inflammatory cytokine produced by T cells [1,2]. Since its initial discovery, MIF is now known to be almost ubiquitously expressed across cell types [3]. MIF has since been implicated in a multitude of innate and adaptive physiologic processes, particularly in response to infection and inflammation, and understanding of its role in tumorigenesis has expanded in recent years [4,5,6]. D-dopachrome tautomerase (DDT), a close homolog of MIF, has also been studied in a variety of inflammatory and neoplastic processes, albeit less than MIF. With increasing studies demonstrating that MIF and DDT enhance immunosuppressive and pro-tumorigenic phenotypes, these cytokines have emerged as promising antitumor targets across a variety of tumor types [7].

## 2. MIF and DDT Structure and Regulation

MIF, a 12.5 kDa homotrimeric protein encoded by the MIF gene on chromosome 22q11.23, functions intracellularly or can be stored in cytoplasmic vesicles, where it is secreted in response to signals such as LPS, TNFα, and hypoxia via upstream TLR and HIF1α signaling [3,8]. Upon secretion, MIF has the ability to operate under autocrine and paracrine signaling. Interaction with its canonical receptor CD74 and non-canonical receptors CXCR2, CXCR4, and CXCR7 drives a wide range of inflammatory, autoimmune, and neoplastic processes [3]. MIF/CD74 signaling requires co-activation with glycoprotein CD44, which plays a crucial role in the MIF/CD74 cognate receptor complex, though MIF itself does not bind directly to CD44. The CD44 gene comprises 19 exons, with 10 participating in alternative splicing to generate variants featuring an extended ectodomain structure (e.g., CD44v1–10) [9].

DDT, a homolog encoded approximately 76 kb away from *MIF*, shares 34% amino acid homology and similarly interacts with CD74/CD44, CXCR4, and CXCR7 to drive downstream signaling [10,11]. Modeling of MIF and DDT structures is depicted in Figure 1a. Unlike MIF, DDT lacks the pseudo-ELR domain required for CXCR2 binding and is therefore unable to signal through this receptor [12,13]. An overview of MIF and DDT chemokine interactions with its cognate (CD74) and non-cognate (CXCR2, 4, 7) receptors is outlined in Figure 1c.

MIF and DDT are constitutively and ubiquitously expressed by a variety of immune and non-immune cell types, including but not limited to the those of the brain, kidney, liver, skin, and heart. MIF and DDT have been widely described as being secreted by macrophages and monocytes, in addition to other immune cells such as neutrophils, B and T lymphocytes, and eosinophils, particularly in response to hypoxia, LPS, ROS, IFN-γ, TNF-α, and glucocorticoids [4,14,15]. They are also secreted by endothelial and epithelial cells in response to LPS, TNF-α, and direct injury, playing a role in neutrophil recruitment and extravasation for wound healing [16,17,18]. Additionally, they are secreted by a wide range of central and peripheral neuronal cells such as astrocytes, microglia, and oligodendrocytes in response to LPS and direct injury [4]. In endocrine systems, MIF is co-localized with insulin in intracellular vesicles of pancreatic beta cells and secreted in response to elevated circulating glucose [19]. MIF secretion is also induced by glucocorticoids in pro-inflammatory states such as infection or sepsis and follows a bell-shaped dose–response curve, with secretion being inhibited at very high glucocorticoid levels to protect the host from a potentially life-threatening inflammatory response [20]. A list of cell types and their stimuli impacting MIF and DDT expression is outlined in Appendix A. 

Intracellularly, MIF is regulated at genetic, epigenetic, and transcriptional levels. At the genetic level, polymorphisms located in the MIF promoter are known to drive MIF expression. Polymorphism rs5844572 located at the -794 locus is comprised of five to eight short-tandem repeats of the CATT sequence (CATT_5–8_), with higher repeat numbers correlating with increased MIF expression [21,22] (Figure 1b). The *DDT* gene, in contrast, lacks a similar promotor region structure (Figure 1b) [23]. In vitro and in vivo Acute Lymphoblastic Leukemia (ALL) models reveal that MIF is positively regulated by UHRF1 (ICBP90) and negatively regulated by HBP1, both of which drive MIF overexpression in disease progression by binding to the promotor region. UHRF1 binds specifically to the CATT sequence, whereas HBP1 binds to promotor regions regardless of the presence of CATT-containing sequences [24]. MIF is also regulated by single-nucleotide polymorphism (SNP) rs755622 at the -173 gene locus. Particularly, C/C and G/C genotypes are associated with the progression of a variety of neoplastic diseases such as cervical cancer, ALL, and gastric carcinoma [25,26,27,28,29,30]. This binding site interacts with the transcription factor activator protein 4 to promote MIF transcription and expression, but whether the -173 G/C has an independent action on MIF transcription is unclear, as it is in strong linkage disequilibrium with the -794 CATT microsatellite repeat [31]. MIF also operates under epigenetic regulation, as the -173 G/C SNP is associated with CpG island hypermethylation and silencing of the tumor suppressors p14ARF and p16INK; histone deacetylase inhibition results in further MIF downregulation [32,33].

At the transcriptional level, various non-coding microRNAs (miRNAs) regulate MIF expression by binding to its 3′-untranslated region. Negative regulators of MIF and downregulated miRNAs include miRNA-451 (prostate cancer, neuroblastoma, gastric carcinoma, and hepatocellular carcinoma) [34,35,36,37], miRNA-144 and miRNA-1228 (hepatocellular carcinoma and gastric carcinoma) [37,38], and miRNA-608 (lung adenocarcinoma and glioblastoma) [39,40]. In contrast, MIF is positively regulated by miRNA-451 (colorectal carcinoma) and miRNA-301b (pancreatic carcinoma) [41,42,43]. No similar regulators of DDT are currently known.

## 3. MIF and DDT Functions in Cancer Progression 

Macrophage dysregulation and suppression were first described in the 1970s and have increasingly become targets of interest in oncology [44,45,46]. Though MIF and DDT have been widely described in inflammation and autoimmunity, interest in these cytokines within oncology has also been relatively recent given their roles in driving cancer hallmarks and their overexpression across a variety of cancers (Figure 2) [47,48,49,50,51,52,53,54]. Here, we provide an overview of tumorigenic phenotypes influenced by MIF and DDT (Figure 3).

MIF and DDT promote cell regeneration and proliferation in cancer by activating the ERK1/2, PI3K-Akt, NFκB, and AMPK pathways, which subsequently activate downstream NF-κB/P-TEFb complexes and drive c-Myb transcription [3,55,56,57,58,59,60,61,62,63,64]. Additionally, early studies identified MIF as a negative regulator of p53, and further evidence has since emerged implicating MIF and DDT’s role in tumor suppressor inhibition [65,66,67]. MIF physically interacts with p53 to inhibit transcription-dependent and independent effects on cell cycle arrest and apoptosis, and they perturb Rb/E2F tumor suppressor activity by disrupting the C-terminal binding region of E2F responsible for binding Rb [68,69,70]. MIF and DDT also suppress apoptotic pathways by downregulating pro-apoptotic Fas receptors, Bax, and Caspase-3 and upregulating anti-apoptotic factors BDNF, MAP2, and BCL2 [71,72,73,74]. This action may be a particularly important role for MIF and DDT in the inflammatory pathogenesis of different cancers, where sustained MIF expression by inflammatory cells in a pre-malignant tumor environment would enhance proliferative signals and cell lifespan, create a deficient response to genotoxic damage, and allow for the accumulation of additional oncogenic mutations [75]. MIF drives tumor metabolic re-programming to confer survival in hypoxic environments by inducing anaerobic metabolism via the NF-κB and HIF-1α pathways [76,77,78,79]. De Azevedo et al. studied murine melanoma cell lines subject to hypoxic conditions and observed the dual MIF and DDT antagonist 4-IPP to downregulate lactate dehydrogenase A and generate less lactate [80]. Further studies show HIF-1α stabilization in hypoxic conditions may concurrently activate MIF and upregulate PD-L1 expression, conferring tumor survival in oxygen-deficient intratumoral environments [76,80,81,82,83,84,85].

MIF and DDT are also linked to tumor vascularization. Inhibition of MIF and DDT reduces tumor vascularization and angiogenic markers in various cell and animal models [86,87,88,89,90,91,92]. Studies propose MIF as an upstream regulator of VEGF, impacting JNFK and AP-1 activity, with a similar role likely for DDT [92,93]. In the context of these findings, it is noteworthy that the CD44 component plays a crucial role in promoting angiogenesis and migration. Upon MIF/CD74 binding, CD44 is recruited to the receptor complex to initiate Src-family kinase activation, CD44 alternative exon splicing, and expression of tumor-associated isoforms such as CD44v3–v6, which are implicated in enhancing cellular migration, adhesion, and invasion. Notably, the CD44v3–v6 isoforms drive these processes by promoting increased matrix interaction and creating neodomains for growth factors and matrix metalloproteinases [9]. Accordingly, MIF/CD74 signaling likely contributes to the invasive and tissue-destructive characteristics observed in transformed cells. In vitro, CXCR4 signaling mediates cellular adhesion to fibronectin, angiogenesis, and migration, consistent with later reports of MIF and DDT enhancing metastatic potential via promoting the mesenchymal-to-epithelial transition (EMT) [94,95,96]. Thus, MIF and DDT further enhance morphologies associated with invasion. 

MIF and DDT also drive tumorigenesis by modulating immune populations within the tumor microenvironment (TME) through cytokine-induced signaling. MIF expression is induced by pro-inflammatory cytokines such as TNF-*α*, IL-5, IFN-*γ*, and TGF-*β* and stimulates the secretion of TNF-*α*, IL-1, IL-6, CXCL8/IL-8, and IL-12 from macrophages [97,98,99]. MIF-dependent secretion of these cytokines promotes proliferation, angiogenesis, and EMT [100,101,102,103]. IFN-γ signaling is particularly important in tumor infiltration and functions by polarizing tumor-associated macrophages (TAMs) from an M2-immunosuppressive type to an M1-pro-inflammatory type; however, it also reduces the presence of CD4 and CD8 T cells in the TME via MIF/CD74 signaling [7,61,104,105,106,107]. Additionally, MIF induces differentiation of myeloid-derived suppressor cells (MDSCs) within the TME, further enhancing tumor permissiveness and immune evasion [108]. 

Additionally, MIF drives a stem cell-like phenotype, resulting in tumor dedifferentiation. This is supported by observations of MIF enrichment in human embryonic stem cells (HESCs) [109]. HESCs mainly express CXCR2 and CXCR7; therefore, MIF is thought to maintain “stemness” through non-canonical pathways. There is also evidence to suggest MIF enhances myocardial repair through autophagy-induced survival of human mesenchymal stem cells [110]. DDT has not yet been implicated in driving a stem cell-like phenotype.

## 4. Evidence of MIF and DDT in Cancer

### 4.1. Hematologic Cancers 

MIF is elevated in multiple hematologic malignancies, including lymphoma, Chronic Lymphocytic Leukemia (CLL), Acute Myelogenous Leukemia (AML), and ALL, where it is correlated with adverse outcomes [74,86,111,112,113,114,115]. Increased circulating MIF levels have been detected in CLL, ALL, and AML, with additional bone marrow enrichment in patients with AML. 

GC/CC phenotypes are associated with an increased risk of childhood ALL, particularly among high-risk ALL and B-ALL groups [116]. In vivo, MIF-/- (MIF-KO) transgenic Eμ-TCL1 mouse CLL models exhibit delayed disease onset and improved survival [117]. MIF suppresses PAX5 and DMTF1 tumor-suppressor activity and upregulates TAp63 and VLA-4 integrin, collectively enhancing CLL survival and bone marrow tropism [118,119,120]. Though DDT’s role in hematologic malignancies is unexplored, studies are underway (NCT03918655) to evaluate the prognostic value of MIF during the treatment of FMS-like tyrosine kinase 3 mutated AML [121].

### 4.2. Osteosarcoma 

Elevated levels of MIF are found in tissue and serum samples from patients with osteosarcoma and are correlated with increased tumor size, pulmonary metastases, and worse survival [122]. In xenograft murine models, administration of the MIF/DDT antagonist 4-IPP led to reductions in tumor burden and metastases [55,122]. 4-IPP induces STUB1 E3 ligand-mediated proteasomal degradation of MIF and reduces subsequent osteolytic activity via suppressing osteoclast formation and promoting osteoblast differentiation [55,123]. Though DDT levels were not directly measured in these studies, its involvement can be inferred, as 4-IPP targets both DDT and MIF [124].

### 4.3. Skin Cancers

Early studies of benzo[α]pyrene-induced fibrosarcomas in MIF-KO mice exhibited enhanced p53 activity in vitro and reduced tumor growth in vivo, suggesting MIF drives development of cutaneous fibrosarcomas via p53 suppression [125]. MIF is also a known driver of melanogenesis and functions by inducing keratinocyte secretion of stem cell factors, a process further enhanced by UV-B [126]. MIF and DDT have been described in UVB-induced (but not chemically induced) non-melanoma skin cancer progression, suggesting that MIF and DDT may play less important roles in chemical carcinogenesis [127]. In melanoma, MIF is present across UV- and non-UV-induced melanomas, where enrichment in serum and tumor is correlated with advanced stages, poor survival, and resistance to immune checkpoint inhibition (ICI) [73,80,128,129,130,131,132]. DDT, though less studied in melanoma, is also enriched in melanoma models [133]. In uveal melanoma, MIF overexpression suppresses natural killer (NK) cell activity, thus establishing an immunosuppression [134]. Uveal melanomas trigger the release of MIF-containing exosomes from hepatocytes, which subsequently enhance tumor viability. Given the high concentration of hepatic NK cells, tumor survival is conferred by MIF-mediated evasion of NK-driven cell lysis [129,135,136]. Though little research has described the role of MIF in acral melanoma, it has similarly been hypothesized to drive pathogenesis via the presence of M2-type macrophages in the TME [137].

### 4.4. Head and Neck Cancers

Elevated MIF expression has also been detected in head and neck squamous cell cancers (HNSCC), particularly in the nasopharynx, hypopharynx, larynx, and oral cavity [138,139,140]. ELISA and single-cell transcriptome analysis have revealed elevated MIF levels in patients with HNSCC [141,142,143]. In patients with nasopharyngeal carcinoma, increased MIF tumor expression is a strong prognostic factor for lymph node metastasis and worse survival [144]. The role of MIF in human papillomavirus (HPV)+ and HPV− cancers has been described, but mechanistically, it remains unclear. HPV suppresses p53 and Rb via the E6 and E7 glycoproteins, respectively, and is a known driver of HNSCC that is associated with better prognosis and survival [145]. Kindt et al. observed by immunohistochemical staining that HPV- tissue had higher total MIF levels compared to HPV+ tissue, but HPV+ cells appear to secrete more MIF, as evidenced by E6- and E7-transfected HNSCC lines producing higher levels of MIF. Kindt et al. also observed 4-IPP treatment of HNSCC cell lines reduced proliferation regardless of HPV status, with HPV+ cells exhibiting a higher IC50, suggesting more MIF requiring neutralization [146]. This group subsequently hypothesized that E6 activation of mTOR drives HIF-1α accumulation and subsequent MIF upregulation. The role of DDT, in contrast, has not yet been described in HNSCC.

### 4.5. Lung Cancers 

MIF plays a significant role in non-small-cell lung carcinoma (NSCLC), with several studies implicating similar functions of DDT. Elevated serum MIF levels in NSCLC predict worse overall and progression-free survival, and co-expression with CD74 correlates strongly with enrichment of tumor-associated CXC chemokines and tumor vascularization [147,148]. MIF or DDT knockdown led to reduced in vitro cellular migration and vascular tube formation; combined knockdown had the greatest effect on dampening CXCL8 and VEGF expression [149]. CXCR4 inhibition suppressed NSCLC migration and invasion, suggesting non-canonical signaling drives metastasis [150]. Interestingly, cisplatin-resistant NSCLC cell lines secrete MIF and enhance macrophage polarization [151]; and ionizing radiation frees MIF from complexing with ribosomal protein S3, enabling its downstream pro-tumorigenic activity [152].

In lung squamous cell carcinoma, tumor MIF expression correlates with lymph node metastasis and worse disease-free survival, and in mesothelioma, CD74 tumor enrichment independently predicts improved survival [153,154]. MIF overexpression in lung adenocarcinoma is associated with increased proliferation and migration and the development of multiple primary tumors [153,155,156]. MIF gene overexpression was also recently identified as a component of a unique gene signature in lung adenocarcinoma, conferring a 53% 5-year recurrence-free survival for patients exhibiting the signature [157]. These studies have been confirmed by in vivo models, whereby decreased tumor growth was observed in mutated MIF and MIF-KO mice, as well as with administration of MIF inhibitor SCD-19 [156]. 

### 4.6. Gastrointestinal Cancers

#### 4.6.1. Esophageal and Gastric Cancer

In esophageal squamous cell carcinoma (ESCC), MIF drives cancer progression via Akt activation and GSK3β tumor suppressor inactivation [158,159]. MIF inhibition decreases tumor growth in murine ESCC models, concordant with patient studies correlating MIF serum and tumor enrichment with tumor dedifferentiation, vascular invasion, lymph node metastases, and worse survival [87,160,161]. Patients with ESCC and poor prognosis also exhibit CXCR4 enrichment, suggesting the importance of non-canonical signaling in tumor progression [162]. Additionally, in patients treated with anti-PD-1 therapy, MIF levels are negatively correlated with survival [163]. To date, DDT has not been described in ESCC. 

Elevated MIF tumor detection in gastric cancer also correlate with angiogenesis, lymph node metastasis, and advanced disease [164,165,166]. High-expression MIF CATT_7_ genotypes have been associated with gastritis and gastric cancer in younger patients, suggesting this MIF risk polymorphism may drive early stages of mucosal inflammation and increase the subsequent risk for gastric cancer [167]. MIF is also released by monocytes responding to *Helicobacter pylori* virulence factor CagA and enhances tumorigenesis [168]. Transcriptomic analysis of CagA+ gastric carcinomas revealed MIF secretion in the TME induces TAM polarization, EMT, and suppression of MAPK4 pathways, all of which are correlated with poor prognosis [169]. P53 suppression mediated by ZFPM2-AS1, an antisense RNA strand that negatively regulates MIF, is also a driver of in vitro gastric carcinogenesis [170]. The specific mechanisms by which DDT may drive gastric carcinoma, however, are less understood.

#### 4.6.2. Hepatocellular Carcinoma

MIF also contributes to hepatocellular carcinoma (HCC), as evidenced by the -173 GC/CC SNP correlating with elevated circulating MIF levels and worse prognosis [28]. In vitro, MIF promotes HCC cell survival and is abrogated by anti-CD74 treatment [171]. Hepatocyte-specific MIF-KO and global CD74 KO mice exhibited reduced tumor burden compared to their WT counterparts. Single-cell transcriptome analysis revealed CD36+ HCC-associated fibroblasts secreted MIF through increased intratumor lipid oxidation; the authors hypothesized that this induced a pro-tumor environment via lipid oxidation, which subsequently activates p38 kinase and drives MIF overexpression [172]. In murine models, combined CD36 and PD-L1 inhibition restored an antitumor immune signature in the TME, further validating the role of MIF in cancer progression. DDT, in contrast, has not been described in HCC.

#### 4.6.3. Pancreatic Carcinoma

Elevated MIF levels in pancreatic carcinoma tissue are correlated with worse prognosis, and studies have consistently demonstrated MIF and DDT’s role in disease progression via driving proliferation, invasion, and anti-apoptotic processes [60,68,173,174]. DDT and MIF knockdown in pancreatic cancer in vitro increased p53 expression and reduced proliferation and invasion in vivo [60,68]. MIF was also found to negatively regulate tumor suppressor NR3C2, an orphan nuclear receptor that inhibits EMT and correlates with improved patient survival. MIF suppression of NR3C2 occurs via upregulation of miR-301b, which binds the 3′ untranslated region of NR3C2 and inhibits its activity [44,175,176] MIF is also present in pancreatic cancer-derived exosomes, which drive the expression, recruitment, and differentiation of myeloid-derived suppressor cells in the TME [177,178,179].

#### 4.6.4. Colorectal Carcinoma

MIF has been widely described in colorectal carcinoma (CRC), where elevated serum MIF levels correlate with increased hepatic metastasis and intratumoral macrophage infiltration [89,180,181]. Additionally, the -173 GC/CC polymorphism is linked to tumor de-differentiation and advanced disease, offering potential as a prognostic biomarker for CRC [182]. In vivo models demonstrate reduced tumor burden and increased apoptosis with MIF blockade [183]. MIF enrichment in KRAS-mutated CRC cell lines contributes to aberrant proliferative signaling, highlighting its potential as a target in treatment-resistant cancers [184,185]. DDT also plays a role in CRC progression via COX-2, a key regulator of β-catenin stability and EMT. Li et al. observed that DDT increases JNK signaling via β-catenin-dependent and -independent mechanisms, and its interaction with atractylenolide I (AT1) leads to p53 deacetylation, both of which confer tumor survival and metastasis [186,187]. Thus, DDT appears to play a role in promoting the adenoma–carcinoma sequence, in part by regulating AT1.

### 4.7. Central Nervous System Cancers

Transcriptomic analyses have identified MIF and DDT as negative prognostic factors in patients with neuroblastoma, regardless of MYCN amplification [188,189]. Mice treated with 4-IPP exhibited reduced neuroblastoma growth and improved survival [190]. MSI1, a neural stem cell marker widely expressed in high-grade gliomas, is thought to be correlated with MIF and drives immunosuppression [106,188]. Furthermore, brain-derived neurotrophic factor conferred neuronal protection in hypoxic and hypoglycemic environments via MIF-dependent apoptotic suppression, consistent with MIF’s known role in enhancing hypoxic survival [71,191]. Of note, the MIF/CXCR4 signaling axis has been implicated in the survival, invasion, and drug resistance of patient-derived neuroblastoma cells in the bone marrow microenvironment and may provide an explanation for the high propensity of bone metastasis in neuroblastoma [190].

MIF and DDT play a pivotal role in aggressive glioblastoma multiforme (GBM) [83,192,193]. GBM cell lines and patient tissue show overexpression of MIF and DDT; levels are correlated with tumor recurrence and poor prognosis [94,194]. Overexpression of CD74/CD44, CXCR2, and CXCR4 in malignant GBM is associated with poor patient prognosis, suggesting that both canonical and non-canonical MIF-dependent pathways contribute to GBM progression [195]. Targeting MIF/DDT pathways offers a therapeutic approach in treatment-resistant GBM, but results are conflicting. 4-IPP combined with radiation therapy reduces in vitro proliferation substantially more than 4-IPP monotherapy, and MIF enrichment is correlated with improved survival in patients treated with neoadjuvant therapy [58,193]. Additionally, MIF in GBM induces immune evasion through MDSCs and modulation of CD8 T cell activity within the TME [108,196].

Thus, MIF targeting has a potential role as an adjunctive therapy with standard treatments. Conversely, bevacizumab resistance appears to be associated with reduced MIF and increased M2-type macrophages, presumably through MIF’s role as an upstream regulator of VEGF production [93,197]. Further research is needed to evaluate these disparate MIF functions and highlight the optimal approach to leveraging MIF-based therapies in treatment-resistant GBM.

### 4.8. Urogenital Cancers

#### 4.8.1. Bladder Cancer

MIF and DDT overexpression has been observed in TCGA analysis of bladder cancers [198,199]. Similarly, CD74 overexpression is detected in high-grade invasive bladder cancer and is associated with proliferation, invasion, and angiogenesis of HT-1376 bladder cancer cells [200,201,202]. MIF-KO murine bladder cancer models demonstrated decreased vascularization and tumor stage [91]. 4-IPP treatment in murine bladder cancer models resulted in decreased tumor weights in MIF-KO versus WT mice, suggesting an additive contribution of DDT inhibition [198]. In bladder cancer cells, MIF knockdown produced a dose-dependent reduction in growth [203]. CPSI-1306, which inhibits the enzymatic region of MIF, reduced cellular proliferation and VEGF expression in vitro and reduced tumor growth and neovascularization in vivo [88].

#### 4.8.2. Prostate Cancer

MIF promotor polymorphisms are associated with worse survival in prostate cancer, with the -173 G/C SNP correlating with increased disease incidence and -794 CATT_7_ correlating with an increased 5-year recurrence risk [204]. This coincides with observations of MIF and CXCR7 overexpression in prostate cancer tissues and in vitro models, including castration-resistant prostate cancers (CRPC) [57,205,206,207]. CXCR7 is vital for the growth and migration of CRPC cell lines, promoting enhanced growth and metastasis in mice. Low miR-451 expression independently predicts worse disease-free and overall survival in CRPC; conversely, enhanced expression in prostate cancer cell lines reduces growth, migration, and invasiveness, negatively correlating with MIF [34]. These findings highlight MIF as a potential target for treating CRPCs. DDT has not been evaluated in prostate cancer.

#### 4.8.3. Renal Cell Carcinoma

High MIF and DDT expression in the kidney impacts renal cell carcinoma (RCC) progression through interactions with HIF1α and HIF2α [90,202]. Hypoxia is a well-known inducer of MIF expression. Similarly, VHL knockout in vitro also decreased DDT levels under hypoxic conditions. DDT and MIF knockdown reduced murine RCC growth, with DDT knockdown producing the most dramatic reduction [90]. This finding suggests a greater contribution of DDT to tumorigenesis in this model. Combined DDT and MIF knockdown exhibited the largest reduction in tumor growth and angiogenesis, further highlighting their synergistic roles and the potential for dual blockade in antitumor treatment [90].

### 4.9. Breast Cancer

Elevated sera and tumor MIF and DDT levels have been observed in breast cancer, validated by cell lines, patient samples and murine models, and are correlated with worse survival [133,208,209]. MIF induces HMGB1 secretion from tumor cells, which subsequently binds TLR4 and activates NF-κB-mediated cell migration [210]. Interestingly, cytosolic MIF correlates with improved survival, suggesting a potential protective role when localized intracellularly [211]. Elevated MIF also has been found in triple-negative breast cancer (TNBC), a highly aggressive breast cancer subtype [212]. In vivo models treated with the small molecule antagonist CPSI-1306 exhibited reduced tumor apoptosis, tumor growth, and metastasis, suggesting MIF to be a potential treatment target in TNBC [211,212]. Likewise, TNBC cells implanted into MIF-KO mice had reduced tumor growth [212]. These findings highlight the role of MIF and DDT in breast cancers.

### 4.10. Gynecologic Cancers

#### 4.10.1. Endometrial Carcinoma

The role of MIF in endometrial carcinoma is less clear. Unlike other cancers, MIF tumor enrichment in endometrial carcinomas is correlated with lower metastatic potential given lower histological grade and lympho-vascular invasion compared to healthy tissue [213,214]. Conflicting studies also suggest MIF tissue overexpression drives carcinoma progression [215]. This is evidenced by MIF upregulation of αv and β3 integrin and VEGF expression in vitro, findings later confirmed in patient-derived endometrial adenocarcinoma tissues [214,216]. Further studies reported MIF secreted by endometrial cancer-associated fibroblasts contributes to immunosuppression [217]. Further research is needed to clarify the role of MIF in endometrial carcinoma progression. To date, there is no research on the role of DDT on endometrial carcinoma.

#### 4.10.2. Cervical Cancer

Increased levels of tumor-expressing and circulating MIF, DDT, and CD74 have been detected in early- and late-stage cervical cancer and have been linked to lymphatic metastasis and up-regulation of E-cadherin and vimentin [60,218,219,220]. Reduced proliferation, vascularization, and migration occurred with MIF and DDT knockdown, as well as with administration of MIF and DDT small-molecule inhibitor ISO-1 [60,218]. Dual blockade of MIF and DDT demonstrated the most profound effect. Murine studies corroborated these results, with dual MIF and DDT inhibition exhibiting the most substantial reduction in tumor size, likely due to greater suppression of involved signaling pathways (Figure 1c) [60].

## 5. Current Therapeutic Applications and Clinical Trials

Despite strong evidence regarding MIF and DDT in tumorigenesis, only a handful of clinical trials have been conducted targeting MIF in oncology, with none targeting DDT. A summary of agents targeting the MIF/DDT/CD74 axis evaluated in vivo and in clinical trials is outlined in Appendix A. Imalumab is a human recombinant antibody targeted against a MIF epitope associated with its oxidation (oxMIF), which may arise in oxidative inflammatory environments, offering a more selective target that spares MIF functions in normal physiology [221,222]. Of note, DDT lacks the CXXC motif crucial for MIF’s redox sensing and may not be targeted by Imalumab [12,13]. In the Phase I trial NCT01765790, the safety of Imalumab was tested in patients with advanced solid tumors and exhibited successful tumor penetration and activation of apoptotic pathways. Encouragingly, the administered regimens did not reach a maximally tolerated dose, indicating a favorable safety profile for patients. This study was halted early due to poor efficacy data and limited enrollment, as 50 of the 69 patients enrolled discontinued treatment due to disease progression or consent withdrawal. Although this study did not extend beyond Phase I, preliminary data suggest a treatment benefit in some patients, with responses primarily manifesting as stable disease for at least 4 months [44,222,223]. A Phase I/IIa trial of Imalumab was conducted in patients with malignant ascites of ovarian cancer; however, it was terminated due to poor study design and limited patient recruitment [223]. A phase IIa study investigating Imalumab in conjunction with fluorouracil/leucovorin or pantimumab versus standard of care in patients with metastatic colorectal carcinoma was also initiated in 2015 but was terminated soon after an early review of safety and efficacy data (NCT01765790) [223].

There are two plausible hypotheses for the limited clinical efficacy to date for Imalumab. First is the uncertainty regarding the relative pro-tumorigenic activities of MIF and oxMIF, as oxidative protein modifications generally reduce productive receptor signal transduction. Second is the likely coordinate expression of DDT in cancers, which is not targeted by Imalumab.

Ibudilast is both an inhibitor of PDE_2_ and allosteric MIF antagonist that has shown efficacy in inducing cell cycle arrest and apoptosis in patient-derived glioblastoma cell lines, with clinical evidence for good CNS penetration [224]. Ibudlilast is already approved in Japan for treating asthma and cerebrovascular disease, with past studies demonstrating the oral form is well tolerated in healthy adults [224]. In a recent Phase II Trial of Ibudilast in Multiple Sclerosis patients, Ibudliast administration slowed whole-brain atrophy, suggesting robust localization and targeting of oligodendrocytes (NCT01982842) [225]. As these cells are also sources of MIF and DDT expression, Ibudliast offers promise as a pharmacologic intervention to CNS cancers such as glioblastoma and brain metastases. It should be noted that the study observed high rates of gastrointestinal side effects associated with the treatment (vomiting, diarrhea, nausea). Additionally, CNS penetration poses a risk for CNS-related toxicities, though it is promising that these side effects were not observed in the trial. Further clinical evidence is needed to confirm the efficacy and toxicity profile associated with the optimal therapeutic window. Indeed, Ibudilast is currently under investigation in combination with temozolomide for newly diagnosed and recurrent glioblastoma patients (NCT03782415) [44,226].

Milatuzumab, an anti-CD74 antibody, was evaluated in a Phase I study for relapsed or refractory Multiple Myeloma (NCT00421525), where it showed disease stability in 26% of patients, along with improved B cell concentration in the serum, without dose-limiting toxicity or rapid drug clearance [227]. In a Phase I/II trial for relapsed or refractory CLL (NCT00868478), Milatuzumab demonstrated a modest overall response, reducing spleen size, enhancing malignant apoptosis, and improving WBC counts, leading to stable disease. Another Phase I–II clinical trial assessed Milatuzumab in frail patients with refractory CLL and detected an improvement in patient quality of life and performance or functional status among patients treated with Milatuzumab [228]. Limitations to this study include the small sample size (*n = 8*) as well as overrepresentation of pre-treated patients, both of which make it difficult to evaluate the generalizability of these results. Though Milatuzumab does not directly neutralize MIF or DDT, these clinical data further support the hypothesis that CD74/MIF/DDT blockade is a promising therapeutic.

Indeed, clinical evidence evaluating direct MIF blockade remains scant, and it has never been tested for DDT antagonism, either alone or in combination with MIF. Moreover, as MIF and DDT exert functions both intra- and extracellularly, the optimal method for MIF and DDT blockade remains unclear. Still, the existing clinical and preclinical data published make a strong case for development of an antitumor therapeutic targeting the MIF and DDT pathway. Overall, targeting this pathway requires a nuanced approach in targeting inflammatory pathways enabling tumor permissiveness without disrupting antitumor inflammatory processes to strike the most therapeutic balance of maximizing clinical outcomes and minimizing immune-related toxic events.

## 6. Future Directions

MIF/DDT/CD74 are potential biomarkers and therapeutic targets across a variety of cancers. Determining which targets within this pathway to neutralize requires further research, especially with respect to determining the distinct versus overlapping functions of MIF and DDT with their canonical versus non-canonical receptors. Given the substantial evidence for MIF and DDT in the progression of a variety of tumors, these are highly tractable targets in oncology with potentially tumor-agnostic applications. Furthermore, MIF and DDT may contribute to ICI resistance in melanoma and pancreatic cancer through T cell suppression and exhaustion, emphasizing the need to target these cytokines and restore antitumor immunity in the TME [229]. Thus, targeting MIF and DDT holds promise for improving ICI response in these highly lethal cancers [230]. A dearth of clinical trial evidence remains a significant limitation to understanding how these agents may work in patients. Given substantial data for roles for MIF and DDT in preclinical cancer models, we encourage continued exploration of clinical and patient-derived models to further evaluate MIF and DDT as effective antitumor targets. Additionally, a human-compatible DDT inhibitor as well as a dual MIF and DDT inhibitor have yet to be developed and studied in preclinical models to validate the hypothesis of targeting MIF and DDT as a synergistic therapeutic approach to cancer in patients.

Targeting MIF and DDT as an antitumor intervention may continue to pose challenges in specificity, as they are ubiquitously and constitutively expressed. Therefore, continued research needs to be performed to optimize pharmacokinetic parameters such as drug stability, delivery to the tumor site, pharmacokinetic development, and binding interactions within the target site. We recommend proceeding with clinical trials of anti-MIF and -DDT agents, with an emphasis on cancer types supported by strong preclinical and clinical evidence for MIF and DDT involvement, such as melanoma. As clinical trials to date have demonstrated overall favorable safety profiles for MIF pathway-targeting agents, we are hopeful that expanding clinical research in MIF and DDT will aid further understanding of these molecules as antitumor targets.

## 7. Discussion

MIF and DDT play pivotal roles in multiple facets of cancer progression and possibly initiation [75,125]. Initially recognized as proinflammatory cytokines, their implication in driving numerous cancer hallmarks has positioned them as potential biomarkers for prognosis, surveillance, and treatment. Extensive experimental and computational research has unveiled intricate signaling mechanisms in MIF/DDT/CD74 pathways across diverse cancer types, shedding light on the complexity of canonical and non-canonical signaling processes within the TME. Moreover, various experimental approaches have assessed a multitude of strategies for MIF and DDT inhibition in both in vitro and in vivo models.

Clinical trials exploring MIF targeting in cancer are limited but show promise. Demonstrated efficacy is possibly limited by unopposed DDT signaling, which is also increased in expression in many cancers (Figure 4). Imalumab showed a modest effect with minimal toxicity in early trials with solid cancers and demonstrated potential in GBM. Milatuzumab has demonstrated efficacy and has been approved for the treatment of patients with multiple myeloma and CLL. Despite progress, therapeutic development for MIF remains in its early stages and requires ongoing evaluation in clinical models. Notably, DDT has not yet been evaluated in clinical trials as an antitumor target, though substantial preclinical data suggest that its potential alone or in combination with MIF should be explored.

## 8. Conclusions

Despite the wealth of evidence implicating MIF and DDT in tumorigenesis, further research is necessary to elucidate the disparate and overlapping mechanisms regulating their function in tumorigenesis. Though clinical exploration of these targets in cancer therapy remains limited, further research is needed to identify optimal blocking methods for these pathways. There is substantial evidence to support the pivotal roles of MIF and DDT in cancer progression and highlight their potential as promising and broadly applicable targets in oncology, arguing for the need for expanded research and clinical trials to establish their efficacy in cancer therapy.

## Figures and Tables

**Figure 1 ijms-25-04849-f001:**
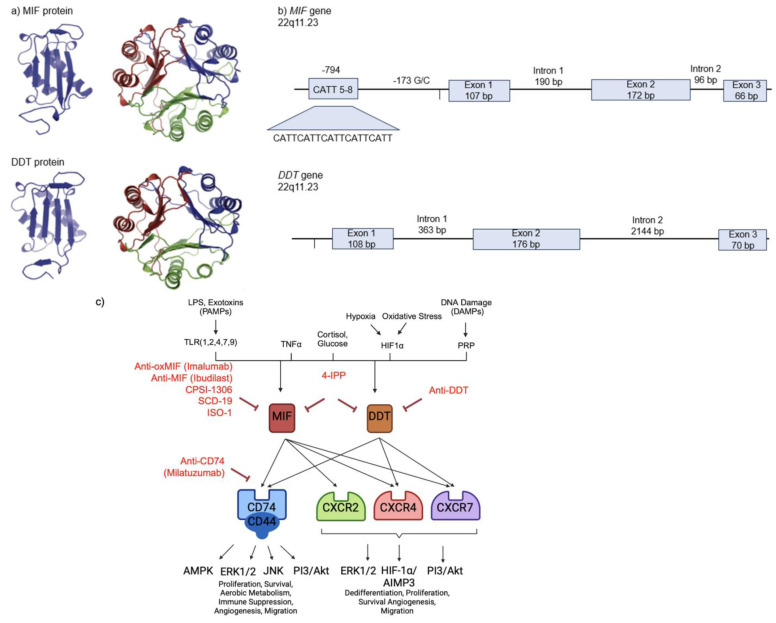
(**a**) Molecular models of MIF and DDT tertiary and homotrimeric structures and (**b**) schematic diagrams of the human *MIF* and *DDT* genes. The *MIF* gene depicts two known promotor polymorphisms: the -794 CATT_5–8_ short-tandem repeat and -173 G/C single-nucleotide polymorphism. (**c**) MIF and DDT binding interactions, subsequent canonical (CD74/CD44) and non-canonical (CXCR2, CXCR4, and CXCR7) pathways, and downstream activities implicated in tumorigenesis. In vivo and therapeutic agents tested in preclinical and clinical cancer models (Imalumab, Ibudilast, Milatuzumab, 4-IPP, ISO-1, CPSI-1306, SCD-19, and anti-MIF and anti-DDT antibodies) are shown at their levels of inhibition in red.

**Figure 2 ijms-25-04849-f002:**
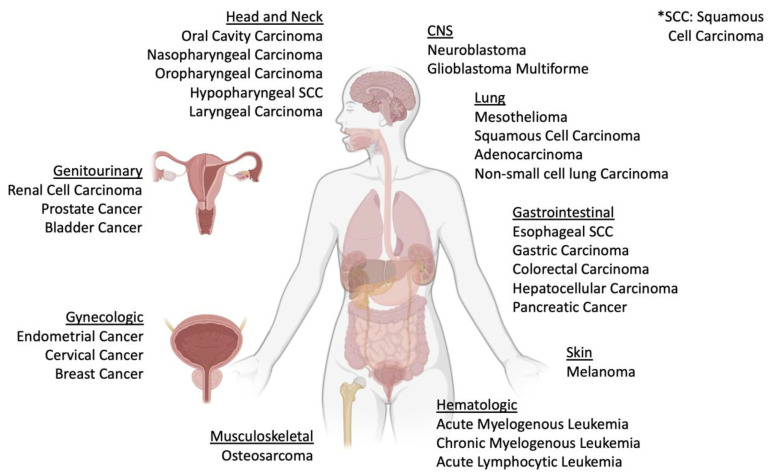
Evidence of MIF and DDT dysregulation has been described in a wide range of cancers, including hematologic, musculoskeletal, skin, head and neck, lung, gastrointestinal, CNS, urogenital, and gynecologic cancers.

**Figure 3 ijms-25-04849-f003:**
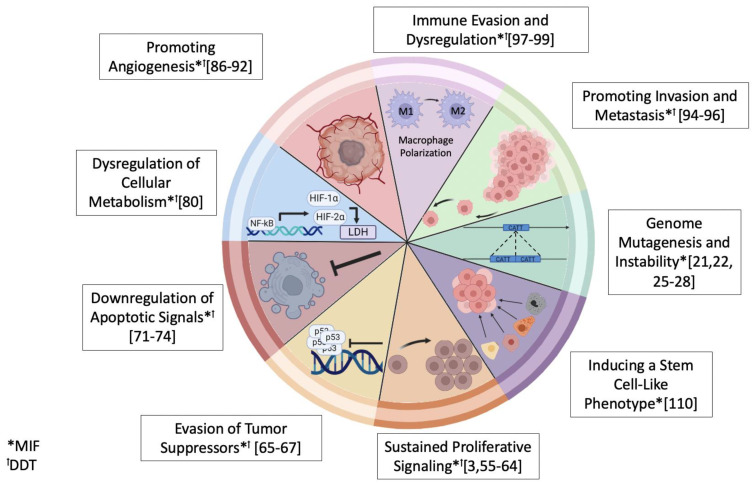
Modified version of cancer hallmarks, as previously described by Hanahan and Weinberg, with evidence of MIF (*) and DDT (^ꝉ^) involvement, accompanied by references.

**Figure 4 ijms-25-04849-f004:**
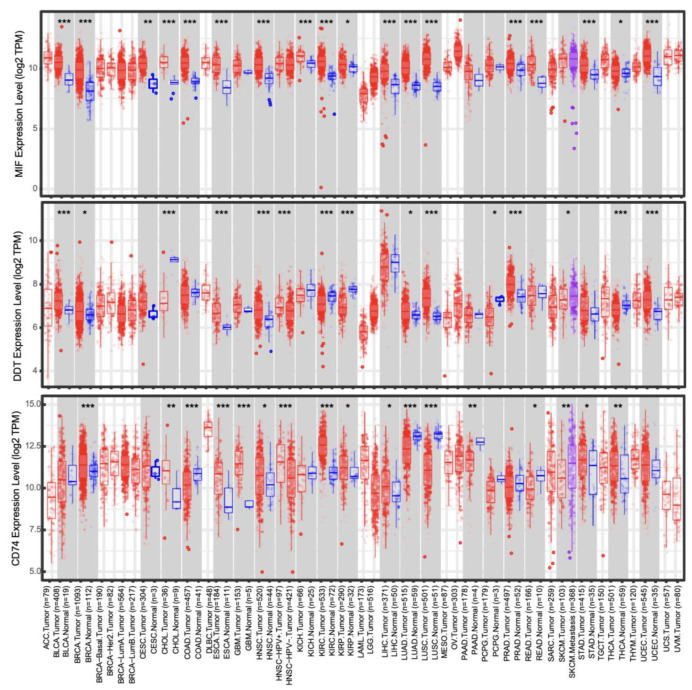
Distributions of MIF (**top**), DDT (**center**), and CD74 (**bottom**) differential expression levels across all TCGA tumors using TIMER 2.0 [52]. Normal (blue), tumor (red), and metastatic (purple) tissues are represented. (Wilcoxon statistical significance *: *p*-value < 0.05; **: *p*-value < 0.01; ***: *p*-value < 0.001).

## Data Availability

The original contributions presented in the study are included in the article/Appendix A, further inquiries can be directed to the corresponding author/s.

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
