# Peer review of "Macrophage Migration Inhibitory Factor (MIF) and D-Dopachrome Tautomerase (DDT): Pathways to Tumorigenesis and Therapeutic Opportunities"

_ijms, 2024, doi:10.3390/ijms25094849_

Round 1
Reviewer 1 Report
Comments and Suggestions for Authors
The paper explores the roles of Macrophage Migration Inhibitory Factor (MIF) and D-Dopachrome Tautomerase (DDT) in cancer progression, detailing their structures, regulatory mechanisms, and therapeutic potential. It suggests that targeting these pathways could offer new antitumor strategies. While the paper discusses clinical trials, there is a missed opportunity to critically analyze why past trials targeting MIF or DDT have not progressed beyond early phases. The author could also offer deeper insights into how these failed clinical trials might influence future research directions and clinical applications. Enhancing these sections with more critical evaluations, and forward-looking perspectives would make the review more valuable to readers
Some minor issues should also be addressed:
-
Line 130: The statement that IFN-γ signaling results in the polarization of tumor-associated macrophages to an M2-immunosuppressive type is wrong. IFN-γ is a cytokine that promotes the polarization of macrophages to the M1 pro-inflammatory phenotype.
-
It should be clarified that the primary MOA of Milatuzumab in NCT00421525 and NCT00868478 is not directly on MIF. Milatuzumab binds to CD74 to induce direct apoptosis or programmed cell death of target cells, and mediates ADCC.
Author Response
The paper explores the roles of Macrophage Migration Inhibitory Factor (MIF) and D-Dopachrome Tautomerase (DDT) in cancer progression, detailing their structures, regulatory mechanisms, and therapeutic potential. It suggests that targeting these pathways could offer new antitumor strategies. While the paper discusses clinical trials, there is a missed opportunity to critically analyze why past trials targeting MIF or DDT have not progressed beyond early phases. The author could also offer deeper insights into how these failed clinical trials might influence future research directions and clinical applications. Enhancing these sections with more critical evaluations, and forward-looking perspectives would make the review more valuable to readers
Response: Thank you for the reviewer’s suggestions. We have expanded on the discussion why past clinical trials have not progressed beyond early phases, and the next steps that should be taken to understand these cytokines as anti-tumor targets. A majority of these changes were addressed in the Current Applications and Clinical Trials section and Future Directions sections. All changes to the manuscript are made in red. We have also expanded on how these failed trials influence future research directions and clinical evaluations. Overall; we have developed these sections to have a more future-oriented perspective.
Line 130: The statement that IFN-γ signaling results in the polarization of tumor-associated macrophages to an M2-immunosuppressive type is wrong. IFN-γ is a cytokine that promotes the polarization of macrophages to the M1 pro-inflammatory phenotype.
Response: Thank you for the reviewer’s invaluable comment. We have corrected the statement in Line 130 which previously stated that IFN-γ induces a polarization of tumor-associated macrophages to an M2 type. We have corrected this.
It should be clarified that the primary MOA of Milatuzumab in NCT00421525 and NCT00868478 is not directly on MIF. Milatuzumab binds to CD74 to induce direct apoptosis or programmed cell death of target cells, and mediates ADCC.
Response: Thank you for the reviewer’s suggestion. We have clarified within the text that Milatuzumab is an anti-CD74 agent; not an anti-MIF or DDT agent, with the following statement: “While Milatuzumab does not directly neutralize MIF or DDT, these clinical data further support the hypothesis that CD74/MIF/DDT blockade is a promising therapeutic.
Please see the attachment for the edited manuscript.

Reviewer 2 Report
Comments and Suggestions for Authors
The review article by Valdez et al. discusses the current knowledge of MIF and DDT in cancer. The authors describe their structure, expression, role in different cancer entities, and therapeutic opportunities. While the manuscript is well-written and supported by four illustrations, it lacks summarized tables and does not cover several important aspects of MIF and DDT biology. Below are constructive comments to guide the authors in enhancing the quality and impact of their review manuscript.
1. Downstream pathways are presented in Figure 1; however, upstream pathways are not highlighted. Please include in the figure how expression and secretion of MIF and DDT are regulated, along with their binding partners.
2. The manuscript lacks a chapter on which cells express and secrete MIF and DDT, and which cells respond to these factors. The chapter on "MIF and DDT Structure and Regulation" contains information that MIF is secreted in response to stimuli such as LPS, TNFα, and hypoxia. However, it is not clear which cells respond to these stimuli. Perhaps the authors could create a table or additional figure to address this and also discuss in the text in a separate chapter.
3. Please create a table to list processes regulated by MIF and DDT with key references. Alternatively, add key references to Figure 3.
4. Figure 1 shows drugs against MIF and DDT, and Chapter 5 also discusses Current Therapeutic Applications and Clinical Trials. Why not summarize drugs and clinical trials in the table?
5. Line 45 – Start a new paragraph on DDT.
Author Response
Please see the attachment for Reply to Reviewer 1 for edited manuscript. All in-text changes have been made in red.
Please see attachment here (Reply to Reviewer 2) for New Supplemental Tables (1-3).
The review article by Valdez et al. discusses the current knowledge of MIF and DDT in cancer. The authors describe their structure, expression, role in different cancer entities, and therapeutic opportunities. While the manuscript is well-written and supported by four illustrations, it lacks summarized tables and does not cover several important aspects of MIF and DDT biology. Below are constructive comments to guide the authors in enhancing the quality and impact of their review manuscript. Downstream pathways are presented in Figure 1; however, upstream pathways are not highlighted. Please include in the figure how expression and secretion of MIF and DDT are regulated, along with their binding partners.
Response: Thank you for this suggestion. We have expanded on the discussion of upstream signaling pathways inducing MIF and DDT expression, as well as included this information in Figure 1. These edits have been made in the body of the manuscript.
The manuscript lacks a chapter on which cells express and secrete MIF and DDT, and which cells respond to these factors. The chapter on "MIF and DDT Structure and Regulation" contains information that MIF is secreted in response to stimuli such as LPS, TNFα, and hypoxia. However, it is not clear which cells respond to these stimuli. Perhaps the authors could create a table or additional figure to address this and also discuss in the text in a separate chapter.
Response: Thank you for this comment. We have expanded the discussion on which cell types secrete MIF and DDT as well as their triggers and external stimuli. As suggested, we also created a new Figure outlining cell types, which has been included as Supplemental Table 1.
Please create a table to list processes regulated by MIF and DDT with key references. Alternatively, add key references to Figure 3.
Response: Thank you for this suggestion. We have included key references in Figure 3, which is included in the body of the edited manuscript.
Figure 1 shows drugs against MIF and DDT, and Chapter 5 also discusses Current Therapeutic Applications and Clinical Trials. Why not summarize drugs and clinical trials in the table?
Response: Thank you for this suggestion. We have included 2 Supplemental tables (Supplement 2 and 3) listing MIF/DDT/CD74 blocking agents in preclinical as well as clinical trials to address this comment. We hope that this table provides an organized framework for recognizing the various approaches researchers have taken in targeting MIF and DDT in cancer, and summarizing their results.
Line 45 – Start a new paragraph on DDT.
Response: Thank you for this suggestion to help with the organization of the review. We have started a new paragraph at Line 45 for the section on DDT’s structure.

Round 2
Reviewer 2 Report
Comments and Suggestions for Authors
The manuscript has been improved. Personally, as a reader, I would like to see the table with clinical trial data included in the main text. However, I will leave it to the authors to decide. The manuscript can be accepted for publication.